# Community purchases of antimicrobials during the COVID-19 pandemic in Uganda: An increased risk for antimicrobial resistance

Agnes N. Kiragga[1,2]*, Leticia Najjemba[1], Ronald Galiwango[1,3,4], Grace Banturaki[1], Grace Munyiwra[5], Idd Iwumbwe[6], James Atwine[6], Cedric Ssendiwala[7], Anthony Natif[7], Damalie Nakanjako[8]

**1** Infectious Diseases Institute, College of Health Sciences, Makerere University, Kampala, Uganda, **2** African Population and Health Research Center, Nairobi, Kenya, **3** African Center of Excellence in Bioinformatics and Data Intensive Sciences, Infectious Diseases Institute, Makerere University, Kampala, Uganda, **4** Center for Computational Biology, Uganda Christian University, Mukono, Uganda, **5** Vine Pharmaceuticals, Kampala, Uganda, **6** Ecopharm Pharmacy, Kampala, Uganda, **7** Guardian Health Uganda, Kampala, Uganda, **8** College of Health Sciences, Makerere University, Kampala, Uganda

* akiragga@aphrc.org

**Data Availability Statement:** The data used in the analysis have been made available and shared as a CSV file.

## Abstract

Self-Medication (SM) involves the utilization of medicines to treat self-recognized symptoms or diseases without consultation and the irrational use of over-the-counter drugs. During the COVID-19 pandemic, the lack of definitive treatment led to increased SM. We aimed to estimate the extent of SM for drugs used to treat COVID-19 symptoms by collecting data from pharmacy sale records. The study was conducted in Kampala, Uganda, where we extracted data from community pharmacies with functional Electronic Health Records between January 2018 and October 2021 to enable a comparison of pre-and post-COVID-19. The data included the number of clients purchasing the following drugs used to treat COVID-19 and its symptoms: Antibiotics included Azithromycin, Erythromycin, and Ciprofloxacin; Supplements included Zinc and vitamin C, while Corticosteroids included dexamethasone. A negative binomial model was used to estimate the incident rate ratios for each drug to compare the effect of COVID-19 on SM. In the pre- COVID-19 period (1st January 2018 to 11th March 2020), 19,285 customers purchased antibiotics which included; Azithromycin (n = 6077), Ciprofloxacin (n = 6066) and Erythromycin (n = 997); health supplements including Vitamin C (430) and Zinc (n = 138); and Corticosteroid including Dexamethasone (n = 5577). During the COVID-19 pandemic (from 15th March 2020 to the data extraction date in October 2021), we observed a 99% increase in clients purchasing the same drugs. The number of clients purchasing Azithromycin increased by 19.7% to 279, Ciprofloxacin reduced by 58.8% to 96 clients, and those buying Erythromycin similarly reduced by 35.8% to 492 clients. In comparison, there were increases of 170%, 181%, and 377% for Vitamin C, Zinc, and Dexamethasone, respectively. The COVID-19 pandemic underscored the extent of SM in Uganda. We recommend future studies with a representation of data from pharmacies located in rural and urban areas to further study pandemics' effect on antimicrobials prescriptions, including obtaining pharmacists' perspectives using mixed methods approaches.

**Funding:** This work was carried out with the aid of a grant from the International Development Research Centre, Ottawa, Canada (ANK, RG). The funders had no role in study design, data collection and analysis, decision to publish, or preparation of the manuscript.

**Competing interests:** The authors have declared that no competing interests exist.

## Introduction

The increasing trend of self-medication and inappropriate drug purchases are becoming major public health concerns [1], driving antimicrobial resistance (AMR). According to the World Health Organization's (WHO) definition, self-medication (SM) is the selection and use of medicines by individuals to treat self-recognized illnesses or symptoms without the consultation of a physician [2]. The most commonly self-prescribed medications are analgesics, antipyretics, antitussives, antidiarrheals, calcium and vitamin supplements, anabolic steroids, sedatives, certain antibiotics, and many herbal and homeopathic remedies [3].

Self-medication is a reasonably widespread practice globally, particularly in deprived communities in low and middle-income countries, and studies have estimated SM in Africa ranging between 30–45% [4–6]. If correctly practiced and under scenarios of proper governing laws on drug prescription, self-medication can self-empower patients to manage their health and improve their health care. In economically deprived communities with overstretched health systems, prescribed and non-prescribed drug purchases may save time for healthcare professionals and resources spent on managing minor and chronic ailments [7]. In addition, drug purchases among community dwellers have been shown to increase survival and reflect changes in healthcare practices [8, 9]. On the contrary, unlimited community drug purchase causes misuse of antimicrobials for viral infections such as colds and influenza [10], which increases the risk of antimicrobial resistance in the respective communities.

In Uganda, the second wave of the SARS-CoV2 Delta Variant led to a spike in COVID-19 cases. Approximately 162,000 cases and 3,557 deaths were reported by April 25, 2022, [11]. During the pandemic's peak, the increasing case fatality rate and the global reports about the disease caused fears that could have led to excessive and rushed purchases of various drugs, including antibiotics, to treat symptoms of COVID-19 disease. Similarly, the desire to receive updated information on the treatment of COVID-19 led to increases in trends in google searches on self-medication for COVID-19 [12] and or self-medication with drugs such as hydroxychloroquine and dexamethasone which may have paused several outcomes, particularly among person with diabetes [13–15]. However, the magnitude to which antimicrobials were used or misused in the community has not been quantified. There remains a dearth of empirical evidence of increased drug purchases and self-medication. A recent study demonstrated an exponential increase in Azithromycin supply during the COVID-19 pandemic but did not provide proof of consumption at the community level [16]. In this study, our objective was to determine the effect of COVID-19 on customer drug purchases in urban Ugandan communities. We mined data from unconventional data sources, such as pharmacy records for purchases of unprescribed drugs as a proxy for community self-medication and drug use during the COVID-19 pandemic. The results of the trends of purchased self-medication before, during, and after the peak COVID-19 pandemic, as presented in this paper, present a potential risk of increased anti-microbial resistance in the respective communities.

## Methods

### Study design and setting

This was a cross-sectional study conducted in Kampala, Uganda, in collaboration with proprietors of private chain pharmacies, including the Ecopharm Pharmacy Uganda Limited and Vine Pharmaceuticals, one of the largest pharmacy chains in Uganda [17, 18]. Pharmacies provide unprescribed drugs and over-the-counter medication to patients with fever, cough, flu, and other common symptoms.

**Data sources.** Electronic drug sale data was obtained from 13 branches of Ecopharm and 35 pharmacies from Vine pharmaceuticals before and after the start of the COVID-19 pandemic in Uganda on 15th March 2020).

## Data variables

We extracted drug sale data between 1st January 2018 to 31st October 2021. The data extracted included monthly customer sales, quantity, and dates of purchases for three main drug classes: antibiotics, supplements, and corticosteroids. These drug classes were purposively selected as they comprised the most common drugs purchased in community pharmacies during the pandemic. Antibiotics included Azithromycin, Erythromycin, and Ciprofloxacin; Supplements included Zinc and vitamin C, while Corticosteroids included dexamethasone. In addition, to relate the drug purchases with the COVID-19 epidemic in Uganda, we extracted daily cases of COVID-19 from the Ugandan Ministry of Health's COVID-19 dashboard [19].

## Data analysis

We determined the proportion of purchasers (clients in the community), the types of drugs (prescribed and non-prescribed) purchases, and the trends in drug purchases before and during the COVID-19 pandemic and during the different waves accompanying SARS-Cov-2 strains. A single group interrupted time-series analysis using segmented regression was used to estimate the effect of the COVID-19 pandemic on drug purchases [18]. The number of clients purchasing selected drugs over time (the outcome) was modeled using a negative binomial distribution for accounting for overdispersion (variance exceeds the mean). The observation time was divided into five segments that matched the period before and after COVID-19, particularly the ascents and descents of Uganda's first and second waves **S1 Fig**. In the model, we estimated five slopes described as slope 1($\beta_1$) (change in drug purchases during the pre-COVID pandemic); slope 2 ($\beta_3$) represents the change in drug purchases during the ascent of the 1st COVID-19 wave; slope 3 ($\beta_5$) represents the change in drug purchases during the descent of the 1st COVID-19 wave; slope 4 ($\beta_5$) represents the change in drug purchases during the ascent of the 2nd COVID-19 wave; and slope 5 ($\beta_7$) represents the change in drug purchases during the descent of the 2nd COVID-19 wave.

## The model equation

The standard Interrupted time series analysis negative binomial regression model assumed the following form:

$$Y_{it} = \beta_0 + \beta_1 T + \beta_2 Z_1 + \beta_3 X_1 + \beta_4 Z_2 + \beta_5 X_2 + \beta_6 Z_3 + \beta_7 X_3 + \beta_8 Z_4 + \beta_9 X_4.$$

$Y_{it}$ is the aggregated outcome variable measured at each equally spaced time point t, and T is a continuous variable indicating the number of days since the start of follow–up. This variable enables us to understand the temporal change in the antibiotic sales pattern and to control for secular trends. $Z_i$'s ($Z_1$, $Z_2$, $Z_3$, $Z_4$) are dummy (indicator) variables representing the first, second, third, and fourth ascent and descent periods (preintervention periods 0, otherwise 1) that happened in March 2020, December 2020, March 2021 and June 2021, respectively. $X_i$'s ($X_1$, $X_2$, $X_3$, $X_4$) are continuous variables counting the number of days since each ascent or descent period in our study (equates to 0 for the pre–intervention (COVID-19) period).

In the case of a single-group method, $\beta_0$ represents the intercept or starting number of clients purchasing the drugs the variable. $\beta_1$ is the slope or trajectory or change in the number of clients purchasing the medications during the pre-COVID-19 era until the start of the

epidemic in Uganda. **$\beta_2$** represents the change in the number of drug purchases during the ascent of the first COVID-19 wave in the period immediately following the introduction of the ascent of the first COVID-19 wave. $\beta_3$ represents the difference in the trend after the increasing peak of the ascent of the first COVID-19 wave. $\beta_4$ represents the change in the level of the outcome immediately after the descent of the first COVID-19 wave. $\beta_5$ represents the difference in the trend of the ascent of the second COVID-19 wave. $\beta_6$ represents the change in the outcome level immediately after the ascent of the second COVID-19 wave. $\beta_7$ represents the difference in the trend after the rise (ascent) of the second COVID-19 wave. $B_8$ represents the change in the level of the outcome immediately after the descent of the second COVID-19 wave. $\beta_9$ represents the difference in the trend of the decline of the second COVID-19 wave. These terms are displayed in **S2 Fig,** and all data used in the manuscript has been shared as **S1 Data**.

The coefficients (β) (incidence rates) from the negative binomial regression models were transformed into exponents to obtain the incident rate ratios (IRR) presented in the tables. The rates allowed us to compare changes between groups during the various periods. The final model adjusted for seasonality by including a term for the two major rainfall seasons: March to May and September to December. This was done to account for the effect of rainfall seasons on the occurrence of upper respiratory infections in Uganda. All the analysis and data visualization were carried out using the R statistical software version 4.1.1 using the Dplyr package for data management and the Mass package for Negative binomial regression.

### Ethical considerations

The study received local approval from the Joint Clinical Research Center Institutional Review and Ethics Committee and the Uganda National Council of Science and Technology (SIR61ES).

## Results

During the pre-Covid era (1st January 2018 and 11th March 2020), a total of 19,285 customers purchased the selected drugs, which included: the antibiotics Azithromycin (n = 6077), Ciprofloxacin (n = 6066) and Erythromycin (n = 997); health supplements including Vitamin C (430) and Zinc (n = 138); and Corticosteroid including Dexamethasone (n = 5577) **S1 Fig**. After the onset of the COVID-19 era between 15th March 2020 and the data extraction date in October 2021, we observed a 99% increase in the number of clients during 20 months. The number of clients purchasing Azithromycin increased by 19.7% to 279, Ciprofloxacin reduced by 58.8% to 96 clients, and those buying Erythromycin similarly reduced by 35.8% to 492 clients. In comparison, there were increases of 170%, 181%, and 377% for Vitamin C, Zinc, and Dexamethasone, respectively.

### Effect of COVID-19 on drug purchases before and during the COVID-19 era

We estimated the impact of the COVID-19 pandemic on purchases of the selected drugs by calculating the change in the number of customers each month **Fig 1.** The changes were assessed before COVID-19 and along the slopes (ascent and descent) of Uganda's two first epidemic waves (**S2 Fig**).

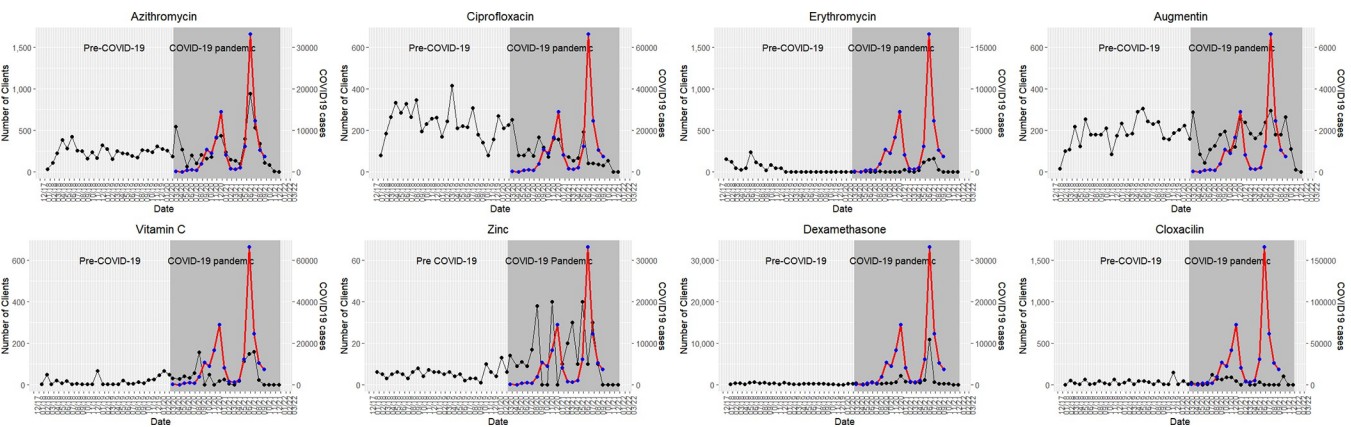

**Fig 1. Shows the trends in purchasing individual drugs pre- and during the COVID-19 pandemic in Uganda (black line).** The red line shows the national monthly number of reported COVID-19 cases (see secondary vertical axis). The time points correspond to the months and years before and during the COVID-19 epidemic in Uganda. For example, March 2020 corresponds to the month in which the first COVID-19 case was diagnosed in Uganda.

## Antibiotics

During the pre-pandemic era, an average of 218.98 customers purchased Azithromycin medications. During the COVID-19 period, however, at the ascent and decline of cases in the first wave, the number of customers purchasing Azithromycin reduced by 79% with a risk ratio (RR) (95% CI) of $\beta_3 = 0.21$ (95% CI: 0.05–0.92) and $\beta_5 = 0.35$ (95% CI: 0.16–0.77). However, towards the peak of the second wave, we observed a 2-fold increase in the number of clients with RR (95% CI) of $\beta_7 = 2.33$ (1.00–5.41), P = 0.015, and as the second wave subsided, the purchase of Azithromycin dropped significantly by 85% with RR (95% CI) $\beta_9 = 0.15$ (0.07–0.30), P<0.001. Before the onset of COVID-19, an average of 206.29 customers purchased Erythromycin medication which reduced gradually by 17% over time, RR (95% CI), $\beta_1 = 0.83$ (0.81–0.84). During the first wave of the COVID-19 era, there were no significant changes in the number of customers buying Erythromycin, as the number of cases increased and decreased. However, during the second wave, as the number of cases peaked, we observed a six-fold increase with RR of $\beta_7 = 6.88$ (95% CI: 3.66–13.43) and later a decline that corresponded with a reducing number of reported COVID-19 cases, RR (95%) $\beta_9 = 0.04$ (0.03–0.07), all P<0.001. A similar trend was observed for Cipfloxacin with no significant changes before COVID-19 and during the first wave, but with an increase as the second wave peaking with RR (95% CI) of $\beta_7 = 2.85$ (1.29–6.32) and the number of purchases reduced by 64% with RR $\beta_9 = 0.36$ (95% CI: 0.20–0.67), as the second wave subsided. The use of Augmentin similarly increased by 43% during the start of the pandemic RR (1.43 (95%: 0.54–3.81), and this later reduced as the second wave declined by 56% RR 95% CI: 0.44 (0.21–0.93) "**Table 1**."

## Supplements

The purchase of supplements, including Zinc and Vitamin C, was affected by the COVID-19 epidemic. Before the onset of COVID-19, there was an average of $\beta_0 = 4.74$ persons purchasing Zinc supplements, which remained relatively stable in the months leading up to the start of the pandemic. However, as the number of cases of COVID-19 decreased during the first wave, the number of clients increased 4-fold with RR of $\beta_5 = 4.84$ (95% CI: 1.35–24.67), P = 0.012. The decreases in the number of clients purchasing Zinc were sustained with a significant reduction at the rise and fall of the second wave of the pandemic with RR (95% CI) of $\beta_7 = 0.22$ (0.04–0.99), P = 0.048 and $\beta_9 = 0.26$ (0.09–0.71), P = 0.011, respectively. We observed remarkable

**Table 1. Multivariable negative binomial regression model for estimating the effect of COVID-19 on antimicrobials and other drug purchases in Uganda.**

| Drug | $\beta_0$ (Constant) (95% CI) | p | B1 (Slope 1) Pre-COVID period RR (95% CI) | p | $B_3$ (Slope 2) Ascent of wave 1 RR (95% CI) | p | $B_5$ (Slope 3) Descent of wave 1 RR (95% CI) | p | $B_7$ (Slope 4) Ascent of wave 2 RR (95% CI) | p | $B_9$ (Slope 5) Descent of wave 2 RR (95% CI) | p |
|---|---|---|---|---|---|---|---|---|---|---|---|---|
| **Antibiotics** | | | | | | | | | | | | |
| Azithromycin | 218.98 (156.59–312.55) | <0.001 | 1.00 (0.98–1.03) | 0.686 | 0.21 (0.05–0.92) | 0.563 | 0.35 (0.16–0.77) | 0.108 | 2.33 (1.00–5.41) | **0.015** | 0.15 (0.07–0.30) | <0.001 |
| Erythromycin | 206.29 (187.31–226.87) | <0.001 | 0.83 (0.80–0.84) | <0.001 | 0.99 (0.62–1.46) | 0.961 | 1.56 (0.87–2.93) | 0.149 | 6.88 (3.66–13.43) | <0.001 | 0.04 (0.03–0.07) | <0.001 |
| Ciprofloxacin | 267.56 (192.80–378.54) | <0.001 | 0.99 (0.97–1.01) | 0.338 | 0.98 (0.89–1.08) | 0.718 | 0.69 (0.39–1.23) | 0.223 | 2.85 (1.29–6.32) | **0.013** | 0.36 (0.20–0.67) | **0.001** |
| Augmentin | 133.04 (88.85–204.90) | <0.001 | 1.02 (1.00–1.05) | 0.068 | 0.95 (0.84–1.07) | 0.454 | 0.87 (0.42–1.77) | 0.688 | 1.43 (0.54–3.81) | 0.462 | 0.44 (0.21–0.93) | **0.025** |
| **Supplements** | | | | | | | | | | | | |
| Zinc | 4.74 (2.65–8.62) | <0.001 | 1.01 (0.97–1.05) | 0.673 | 1.08 (0.91–1.28) | 0.383 | 4.84 (1.35–24.67) | **0.012** | 0.22 (0.04–0.99) | **0.048** | 0.26 (0.09–0.71) | **0.011** |
| Vitamin C | 9.06 (4.26–22.07) | <0.001 | 1.04 (0.99–1.09) | 0.192 | 0.95 (0.63–1.37) | 0.731 | 0.58 (0.06–4.98) | 0.513 | 35.15 (1.65–187.21) | **0.008** | 0.01 (0.00–0.07) | <0.001 |
| **Corticosteroids** | | | | | | | | | | | | |
| Dexamethasone | 413.31 (219.09–840.79) | <0.001 | 0.95 (0.91–0.99) | **0.014** | 1.12 (0.91–1.38) | 0.302 | 0.47 (0.14–1.56) | 0.215 | 3.07 (0.61–15.52) | 0.185 | 0.17 (0.05–0.60) | **0.005** |

increases in the number of persons purchasing Vitamin C during the second wave of the pandemic, with a 35-fold increase from the pre-pandemic era, RR (95% CI) of $\beta_7$ = 35.15 (1.65–187.2), P = 0.008, and the purchases reduced with reducing COVID-19 cases, RR (95% CI) $\beta_9$ = 0.01 (0.00–0.07), P<0.001 "**Table 1**".

## Corticosteroids

We observed many people purchasing Dexamethasone before the pandemic, with an average of β0 = 413.31 monthly customers, which reduced slightly over time. However, during the second wave of COVID-19, there was a non-statistically significant three-fold increase in the number of customers, RR (95% CI) $\beta_7$ = 3.07 (0.61–15.52), and after that, a substantial decrease in purchases at the end of the second wave, $\beta_9$ = 0.17 (0.05–0.60), P = 0.005 "**Table 1**".

## Discussion

We observed an increase in drug purchases and an indicative increase in self-medication during the COVID-19 epidemic in Uganda. There were significant increases in the purchase of antibiotics, including Azithromycin and Erythromycin. Moreover, the increases in the number of drug purchases correlated with the different waves in the country, especially during the second wave of the pandemic, which was driven by the (SARS-CoV-2) B.1.617.2 (delta) variant [20].

In Uganda, like many other parts of the world, the alarming infodemic, rising self-medication, and stockpiling of drugs, including antibiotics, led to elevated purchases of critical medications believed to treat COVID-19 [21, 22]. In addition, the exponential increase in uptake of antibiotics such as Azithromycin for the misinformed treatment of viral infections, including

COVID-19, likely led to heightened antimicrobial resistance [16, 23]. Similarly, we demonstrated significant increases in people purchasing health supplements, including vitamin C.

This could have been influenced by beliefs spread through social media or rumors about the efficacy of particular drugs, particularly during the lockdown, when access to health facilities was reduced. In addition, the increase in Vitamin C was driven by recommendations from scientific evidence, health workers, and social media about its role in increasing immunity and improving the survival of COVID-19 patients [24]. Finally, the self-medication of the selected drugs could have been stimulated by media and influencers who encouraged people in the community to stockpile medicines and supplements, as depicted in our results.

Unfortunately, individuals who self-medicate might misinterpret the recommended dosage and frequency, thus leading to an increased likelihood of severe side effects, drug interactions, and limited efficacy. The unregulated use of corticosteroids such as dexamethasone, which is only recommended among persons with severe conditions related to cytokines, poses a serious risk of severe adverse outcomes among people with underlying comorbidities, including diabetes [25].

Our results demonstrated a significant increase in the use of Azithromycin, particularly during the second wave. This steep increase in the use of antibiotics was similar to what was reported in India [16, 26], a country that continues to lead in drug manufacturing. In India, the increase was noted as early as the first wave, while in Uganda, the growth was higher in the second wave, with the most significant consumption among presumptive or confirmed COVID-19 cases. These findings seriously affect antimicrobial resistance in low- and middle-income countries. During the pandemic, the overstretched health systems, fear of contracting the disease, and non-pharmaceutical prevention measures, such as the suspension of public transport, made it impossible to access treatment and foreclosed the option of self-medication at a nearby community pharmacy [27]. Like many other resource-limited settings, Uganda's access to proper health care was limited during the pandemic. Community pharmacies presented a potential opportunity to access treatment for COVID-19-related symptoms. Community pharmacies offered clients convenience (e.g., short waiting times), privacy, and efficiency, characteristics that made them desirable destinations during the peak of the second wave in Uganda, driven by the Delta variant [28, 29]. In June 2021, during the second wave of the pandemic in Uganda, a widely distributed flyer recommended using a COVID-19 treatment package that included Azithromycin, Augmentin, Dexamethasone, Vitamin C, and Zinc, leading to massive increases in drug purchases.

Proper communication about the harmful effects of irrational drug use is essential during pandemics. More importantly, the community should be sensitized about seeking advice from health experts before purchasing and eventually using any medication. In Uganda, social media was critical in promoting self-medication for COVID-19. The drug stockpiling and use of unprescribed medication revealed the public's vulnerability during the global pandemic threat [30]. The fear of contracting and the experience of being diagnosed with COVID-19 caused members of communities to buy drugs to feel secure and treat COVID-19 symptoms. However, the consequential effect on antimicrobial resistance in the community is yet to be evaluated. Therefore, pharmacists and drug shops had a vital role in the community response to the COVID-19 pandemic. Regular critical analysis of pharmacy purchase data may be relevant to inform appropriate guidance on self-medication in the community [31].

## Strengths and limitations

The major strength of our study is its originality and innovativeness in using customer purchase data. First, due to the need for studies using similar data sources in Africa, this research

could be used as a benchmark for future studies that utilize unconventional data sources such as community pharmacy records. Such unexplored data sources can address essential health outcomes and activities in challenging times, such as pandemics. Secondly, unlike previous studies that compared drug use pre- and post-COVID-19, our unique approach of splitting the post-pandemic era into different waves to correct special drug purchases with the increasing and reducing reported cases of COVID-19 provides additional information on the effect of each of the waves on community self-medication based on the severity of the variants.

Our study was limited to pharmacies with electronic receipting services from which the data could be extracted. Therefore, our data is likely an underestimation of the increase in self-medication during the pandemic because most pharmacies and drug shops in the community still need electronic receipting systems. In Cameroon, mathematical modeling approaches have been used to estimate the effect of self-medication on the COVID-response [32]. Among participants in a Nigerian study, nearly 80% reported purchasing drugs from community pharmacies [6], thus emphasizing the need for innovative ways to estimate self-medication following data mining pharmacy drug records. We postulate that customer purchase data and drug stockpiling during the epidemic are proxy indicators of increased antimicrobial consumption in the absence of measures of direct consumption. We, therefore, recommend interventions to measure antimicrobial consumption and resistance patterns in the community. Our data lacked demographic characteristics of the individuals who purchased the medication and is therefore limited in multivariable modeling of the drivers of drug purchases. The absence of unique identifiers in the pharmacy database made it impossible to identify repeated purchases from a single individual. Therefore any correlation among clients was not accounted for in the modes. With increased advocacy for data sharing and the use of data science methods to mine pharmacy records, the value of such unconventional data sources will go a long way in informing public health responses in our settings of interest. Finally, we are aware that there could be structural differences in clients who purchase drugs from private and public pharmacies that might be driven by their spending ability and differences in drug pricing in rural and urban areas in Africa [33]. Due to the high prices of drugs, the increase in purchases might have been more in urban Uganda than in rural areas [34]. Therefore, we recommend that future studies be undertaken with a larger sample size from pharmacies located in rural and urban areas. Studies should consider pharmacists' perspectives using a mixed methods approach.

## Conclusion

Analysis of drug purchase records pharmacies revealed significant increases in antibiotics and supplements purchases correlated with the trends in COVID-19 cases during the pandemic. The unregulated or unrestricted availability of over-the-counter drugs and their use for the treatment of non-bacterial infections such as SARS-CoV-2 remains of great concern for consequential antimicrobial resistance patterns. More data is needed to understand the impact of COVID-19 self-prescription practices on antimicrobial resistance patterns in the community.

## Supporting information

**S1 Fig. Shows the trends in actual purchases of all drugs before and during the COVID-19 pandemic in Uganda.** The time points correspond to the months and years before and during COVID-19. For example, March 2020 corresponds to the month in which the first COVID-19 case was diagnosed in Uganda.
(TIFF)

**S2 Fig. The black line shows the number of clients purchasing a selected drug before and during the COVID-19 epidemic in Uganda.** The COVID-19 era is divided into four periods based on the ascent and descent of the two COVID-19 waves in Uganda. The red line shows the monthly number of COVID-19 cases (see secondary vertical axis) reported during the first two waves in the country. The blue line illustrates the slopes and descents estimated from the negative binomial regression model and their corresponding slopes and coefficients. (TIFF)

**S1 Data. Data file.**
(CSV)

## Acknowledgments

This work was carried out with a grant from the International Development Research Centre, Ottawa, Canada, and the Swedish International Development Cooperation Agency. However, the views expressed herein do not necessarily represent those of IDRC or its Board of Governors.

## Author Contributions

**Conceptualization:** Agnes N. Kiragga, Ronald Galiwango.

**Data curation:** Leticia Najjemba, Ronald Galiwango, Grace Banturaki, Grace Munyiwra, Idd Iwumbwe, James Atwine, Cedric Ssendiwala, Anthony Natif.

**Formal analysis:** Agnes N. Kiragga, Leticia Najjemba, Ronald Galiwango.

**Methodology:** Agnes N. Kiragga, Leticia Najjemba, Ronald Galiwango.

**Project administration:** Agnes N. Kiragga.

**Supervision:** Agnes N. Kiragga, Damalie Nakanjako.

**Validation:** Agnes N. Kiragga, Ronald Galiwango.

**Visualization:** Agnes N. Kiragga, Leticia Najjemba, Ronald Galiwango.

**Writing – original draft:** Agnes N. Kiragga, Leticia Najjemba, Ronald Galiwango.

**Writing – review & editing:** Agnes N. Kiragga, Ronald Galiwango, Damalie Nakanjako.

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
