## [Decision Letter · Decision Letter 0]

3 Aug 2022

PGPH-D-22-00781

Increased Self-prescribed Community Purchases of antimicrobials during the COVID-19 Pandemic in Uganda: A risk for antimicrobial resistance

Dear Dr. Kiragga,

Thank you for submitting your manuscript to PLOS Global Public Health. After careful consideration, we feel that it has merit but does not fully meet PLOS Global Public Health’s publication criteria as it currently stands. Therefore, we invite you to submit a revised version of the manuscript that addresses the points raised during the review process.

We look forward to receiving your revised manuscript.

Kind regards,

Michele Spinicci

Academic Editor

Journal Requirements:

1. Please amend your detailed online Financial Disclosure statement. This is published with the article. It must therefore be completed in full sentences and contain the exact wording you wish to be published.

2. Please update your online Competing Interests statement. If you have no competing interests to declare, please state: “The authors have declared that no competing interests exist.”

3. Manuscript datafile for journal.csv is currently uploaded as file type “Other”, which is not viewable by the reviewers. Please change the file type(s) to 'Supporting Information' and include a legend in the manuscript.

4. Please provide separate figure files in .tif or .eps format and remove any figures embedded in your manuscript file. Please also ensure that all files are under our size limit of 10MB.

5. We notice that your supplementary figure is included in the manuscript file. Please remove them and upload them with the file type 'Supporting Information'. Please ensure that each Supporting Information file has a legend listed in the manuscript after the references list.

Additional Editor Comments (if provided):

Reviewers' comments:

Reviewer's Responses to Questions

**Comments to the Author**

1. Does this manuscript meet PLOS Global Public Health’s publication criteria? Is the manuscript technically sound, and do the data support the conclusions? The manuscript must describe methodologically and ethically rigorous research with conclusions that are appropriately drawn based on the data presented.

Reviewer #1: Partly

Reviewer #2: Partly

2. Has the statistical analysis been performed appropriately and rigorously?

Reviewer #1: Yes

Reviewer #2: Yes

3. Have the authors made all data underlying the findings in their manuscript fully available (please refer to the Data Availability Statement at the start of the manuscript PDF file)?

Reviewer #1: Yes

Reviewer #2: Yes

4. Is the manuscript presented in an intelligible fashion and written in standard English?

Reviewer #1: Yes

Reviewer #2: Yes

5. Review Comments to the Author

Reviewer #1: The time series model may be adequate, but it is suggested that it better clarify the theory with the results.

Clarify the specific R package they use

The graphs should be better explained. There are black lines and red lines and you don't know which is which. It seems that the results are not taking advantage of the model, because they are calculating RRs and their confidence intervals, but I think that RRs are not obtained in those models.

Figure 1 shows a series with very atypical data that must have an explanation. The unused scale does not show whether this happens for all products or not.

Table 1 is from a negative binomial model that has slices at the time points used by the authors, but it is not clear how they get there. In the final graphs it seems that they forget the time points. In the last figure they show a pre and post covid adjustment, it is not clear if they forget the moments of time and why they do it.

In general, it seems that there is a lack of coherence between the methodological part and the results. I consider that there is a lack of explanation of what is done and what is proposed and how it is explained.

It is an investigation with an adequate methodology, but with inconsistencies between what they propose and what they present. It could be improved how the results are presented and that the tools used and the reference literature coincide throughout the document.

Additional Information

Bibliography reference 16 and 18 are not complete and I consider that they are not necessary. Reference 17 does not seem to correspond to the text.

In the abstract that uses the term data-mining to do the information set. For me it is just a collection of data from different sources and they call “supplements” but in results used "micronutrients"

Reviewer #2: Dear Authors,

I have read with interest your manuscript. The purpose of the study, as well as the methods used are relevant. Additionally, using that kind of data opens the door to future studies on sef-medication, which is a field of research very useful to identify key messages to be used in communication campaigns.

Notwithstanding this, I would like to focus os several aspects which I consider important for the present study:

(1) To me, the most relevant point are the selected medicines. I agree that the medicines included should be there. But what about other medicines that were under the spot because of rumours, social media, etc.?

- The most clear example is "Augmentin" (that you already cite as part of a flyer). "Augmentin" should also be included in that analysis.

- But, even more, it would be highly relevant to observe the behaviour of "Augmentin" as compared to the behabiour of other popular brand names of amoxicillin + clavulanic acid. This help to identify the effects of spreading false information using brand names, for example.

- Hydroxychloroquine and chloroquine were also used, especially during the initial stages of the pandemic. The evolution of the sales of both medicines should also be included.

- The same for ivermectin.

- And medicines used to manage fever, such as paracetamol and ibuprofen.

(2) Another aspect related to medicines is the importance of having a series of "control" medicines (for example, laxatives, popular combinations in Uganda (vitamins, etc.).

(3) My last point refers to the limitations of the study, which should be clearly addressed, as well as the differences between the included pharmacies and other pharmacies of the country, sociodemographic aspects, potential different impact of infodemia to clients going to private pharmacies and clients going to public pharmacies, a discussion on the expected behaviour in the other pharmacies, taking into account previous studies, etc.

Kind regards

6. PLOS authors have the option to publish the peer review history of their article (what does this mean?). If published, this will include your full peer review and any attached files.

**Do you want your identity to be public for this peer review?** For information about this choice, including consent withdrawal, please see our Privacy Policy.

Reviewer #1: **Yes: **José Julián López G.

Reviewer #2: **Yes: **Albert Figueras

---

## [Decision Letter · Decision Letter 1]

2 Jan 2023

PGPH-D-22-00781R1

Increased Self-prescribed Community Purchases of antimicrobials during the COVID-19 Pandemic in Uganda: A risk for antimicrobial resistance

Dear Dr. Kiragga,

Thank you for submitting your manuscript to PLOS Global Public Health. After careful consideration, we feel that it has merit but does not fully meet PLOS Global Public Health’s publication criteria as it currently stands. Therefore, we invite you to submit a revised version of the manuscript that addresses the points raised during the review process.

We look forward to receiving your revised manuscript.

Kind regards,

Muhammad Fawad Rasool

Academic Editor

Journal Requirements:

1. "Manuscript datafile for journal.csv" is currently uploaded as file type “Other”, which is not viewable by the reviewers. Please change the file type(s) to 'Supporting Information' and include a legend in the manuscript if you wish it/them to be included in review.

Additional Editor Comments (if provided):

Reviewers' comments:

Reviewer's Responses to Questions

**Comments to the Author**

1. If the authors have adequately addressed your comments raised in a previous round of review and you feel that this manuscript is now acceptable for publication, you may indicate that here to bypass the “Comments to the Author” section, enter your conflict of interest statement in the “Confidential to Editor” section, and submit your "Accept" recommendation.

Reviewer #3: All comments have been addressed

Reviewer #4: All comments have been addressed

Reviewer #5: All comments have been addressed

2. Does this manuscript meet PLOS Global Public Health’s publication criteria? Is the manuscript technically sound, and do the data support the conclusions? The manuscript must describe methodologically and ethically rigorous research with conclusions that are appropriately drawn based on the data presented.

Reviewer #3: Yes

Reviewer #4: Yes

Reviewer #5: Yes

3. Has the statistical analysis been performed appropriately and rigorously?

Reviewer #3: N/A

Reviewer #4: Yes

Reviewer #5: Yes

4. Have the authors made all data underlying the findings in their manuscript fully available (please refer to the Data Availability Statement at the start of the manuscript PDF file)?

Reviewer #3: Yes

Reviewer #4: Yes

Reviewer #5: Yes

5. Is the manuscript presented in an intelligible fashion and written in standard English?

Reviewer #3: No

Reviewer #4: No

Reviewer #5: Yes

6. Review Comments to the Author

Reviewer #3: Title: is too long.

Methodology: no clear study design.

References: NO.16 & 18 needs to rewrite in a right way (Author, A. (year, month day). Title of article. Title of Newspaper. Retrieved from “home page web address”)

Reviewer #4: Thank you for giving me the opportunity to review this manuscript. Overall, the authors have addressed the comments raised by reviewers. However, minor changes are still required before this manuscript can be deemed suitable for publication.

1. Please improve the language of the manuscript for consistency, grammar and punctuation. For example, use a suitable consistent word for the COVID-19 pandemic everywhere in your paper, and revise any other formatting errors.

2. I think it would be more useful if you include regression model explanation in your Methodology section of the manuscript instead of writing it in the Supplementary figure file. This would be easier for the readers and provide more clarity on your work.

3. The legend for data used is still not present in the manuscript. Please follow PLOS convention for uploading data as Supplementary Information.

4. Table 1 has inconsistencies in representation of Confidence intervals. Please adjust the parentheses.

5. Figure 1 and 2 titles are confusing. Please re-write, for example, if the graphs are from interrupted time series analysis results or actual purchases.

6. You could include Strengths and Limitations with a different heading under Discussion section.

Reviewer #5: Increased Self-prescribed Community Purchases of antimicrobials during the COVID-19 Pandemic in Uganda: A risk for antimicrobial resistance

Abstract

Line number 3, ‘’aimed to’’ deleted, and ‘’d’’ added to estimate

Recommendations was not included in the abstract. This should be included.

Keywords should be included

Introduction

Line number 7 to be deleted. It is repetition of Line number 6.

Line number 8 (‘’Previous studies have estimated the prevalence………….) to be merged with Line number 6

Line number 15 (Need for authors to provided reference for the sentence: ‘’In Uganda, the second wave of the SARS-CoV2 Delta Variant…………by April 25, 2022

Line number 17: Need for authors to include Reference to the ending of the statement from Line number 16 (……., including antibiotics to treat symptoms of COVID-19 disease)

Need for authors to clearly reframe the objectives of the manuscript

Methodology

Change Methodology to Methods

Information under Data variables will be better presented using a flow diagram

Need for operational definitions of some terms under the methods section: prescribed, non-prescribed, pre and post COVID-19 pandemic periods etc

‘’To determine the effect of COVID-19 pandemic on customer drug purchases’’ appears to be part of the study objectives which supposed to be well articulated under the Introduction section

‘’Our hypothesis before the study was that……..created the segments of interest’’ supposed to be under the Introduction section

Data analysis section to have sub-sections for easy and well articulate flow of information

Results

Line number 1 ‘’During the pre-Covid era 11th March 2020’’. This should be captured under the Methods section

Supplements: Table 1 to be included at the end of the sentence under supplements

Corticosteroids: Table 1 to be included at the end of the sentence under corticosteroids

Discussion

Line number 35: ‘’the role of’’ to be deleted from ‘’It therefore clear that…….’’

Tables

Incomplete information provided as title. Need to include information on where and when

Figures

No title. Need to insert complete title that will provide information on ‘’What, Where, and When’’

References

URL to be provided with date accessed most especially for references from WHO website

For references with URL being provided, authors should include date when the information was assessed

7. PLOS authors have the option to publish the peer review history of their article (what does this mean?). If published, this will include your full peer review and any attached files.

**Do you want your identity to be public for this peer review?** For information about this choice, including consent withdrawal, please see our Privacy Policy.

Reviewer #3: **Yes: **Dr. Mohammed Ahmed Abdelrahman

Reviewer #4: **Yes: **Mujahid Abdullah

Reviewer #5: No

<quillbot-extension-portal></quillbot-extension-portal>

---

## [Decision Letter · Decision Letter 2]

17 Jan 2023

Community Purchases of antimicrobials during the COVID-19 Pandemic in Uganda: An increased risk for antimicrobial resistance

PGPH-D-22-00781R2

Dear Dr. Kiragga,

We are pleased to inform you that your manuscript 'Community Purchases of antimicrobials during the COVID-19 Pandemic in Uganda: An increased risk for antimicrobial resistance' has been provisionally accepted for publication in PLOS Global Public Health.

Best regards,

Muhammad Fawad Rasool

Academic Editor

Reviewer Comments (if any, and for reference):

Reviewer's Responses to Questions

**Comments to the Author**

1. If the authors have adequately addressed your comments raised in a previous round of review and you feel that this manuscript is now acceptable for publication, you may indicate that here to bypass the “Comments to the Author” section, enter your conflict of interest statement in the “Confidential to Editor” section, and submit your "Accept" recommendation.

Reviewer #4: All comments have been addressed

Reviewer #5: All comments have been addressed

2. Does this manuscript meet PLOS Global Public Health’s publication criteria? Is the manuscript technically sound, and do the data support the conclusions? The manuscript must describe methodologically and ethically rigorous research with conclusions that are appropriately drawn based on the data presented.

Reviewer #4: Yes

Reviewer #5: Yes

3. Has the statistical analysis been performed appropriately and rigorously?

Reviewer #4: Yes

Reviewer #5: Yes

4. Have the authors made all data underlying the findings in their manuscript fully available (please refer to the Data Availability Statement at the start of the manuscript PDF file)?

Reviewer #4: Yes

Reviewer #5: Yes

5. Is the manuscript presented in an intelligible fashion and written in standard English?

Reviewer #4: Yes

Reviewer #5: Yes

6. Review Comments to the Author

Reviewer #4: The authors have addressed my comments.

Reviewer #5: (No Response)

7. PLOS authors have the option to publish the peer review history of their article (what does this mean?). If published, this will include your full peer review and any attached files.

**Do you want your identity to be public for this peer review?** For information about this choice, including consent withdrawal, please see our Privacy Policy.

Reviewer #4: **Yes: **Mujahid Abdullah

Reviewer #5: No

<quillbot-extension-portal></quillbot-extension-portal>